# Isolation of *Bacillus altitudinis* 5-DSW with Protease Activity from Deep-Sea Mineral Water and Preparation of Functional Active Peptide Fractions from Chia Seeds

**DOI:** 10.3390/microorganisms12102048

**Published:** 2024-10-10

**Authors:** Hao Jin, Eun-Gyo Lee, Faiza Khalid, Seung-Wha Jo, Sang-Ho Baik

**Affiliations:** 1Department of Food Science and Human Nutrition, Jeonbuk National University, Jeonju 54896, Jeonbuk, Republic of Korea; blueberry-73jinhao@naver.com (H.J.); dldmsry6651@naver.com (E.-G.L.); faiza5951@gmail.com (F.K.); 2Microbial Institute for Fermentation Industry (MIFI), Sunchang 56048, Jeonbuk, Republic of Korea; tmdghk606@hanmail.net

**Keywords:** *Bacillus altitudinis*, chia seeds, peptide, proteases, protein

## Abstract

In this study, we successfully isolated *Bacillus* strains with high protease activity from deep-sea mineral water in Korea and used them to obtain functional peptide fractions from chia seeds. The obtained *Bacillus* strains showed a high similarity of 99% with *B. altitude* with a long rod type (named *B. altitudinis* 5-DSW) and high protease activity at 40 °C, and 70% of the activity remained even at 70 °C. The defatted chia seed protein (15–50 kDa) was treated with crude protease from *B. altitudinis* 5-DSW and digested into small peptides below 20 kDa. The obtained chia seed peptides showed 3 times and 1.5 times higher antioxidant activity in DPPH and ABT radical scavenging assays, respectively. Moreover, chia seed peptides showed enhanced AChE inhibitory activity with an IC_50_ value of 14.48 ± 0.88 μg/mL and BChE inhibition activity with an IC_50_ value of 10.90 ± 0.80 μg/mL. Our results indicate that the newly isolated *B. altitudinis* 5-DSW and chia seed protein hydrolysates have potential applications in biotechnology and functional food development, enhancing the nutritional quality and value-added utilization of chia byproducts.

## 1. Introduction

*Bacillus altitudinis* is a Gram-positive, rod-shaped bacterium classified in the phylum Firmicutes, which is usually isolated from diverse cryogenic environments, such as the ocean [1], deep freshwater [2], soil [3], and silt [4]. *B. altitudinis* has received great attention owing to its industrial applications, including high-level bioremediation of various metals [5], lignins [6], agricultural bio-fertilizers [7], agro-waste bioconversion of lignocellulosic (LC) plant matter, and probiotics with antimicrobial activity [8]. Recently, it was revealed that the extracellular polysaccharides produced by *B. altitudinis* may be strongly related to the viability of these strains in harsh deep-sea ecosystems. In addition, the diverse spectra of enzymes such as cellulase [9], xylanase [10], acid-stable catalases [11], uricase [12], and proteases [13] have gained special interest because of their specific properties as candidates for chemical catalysts. In particular, proteases (EC 3.4.21-24) derived from alkaliphiles of *B. altitudinis* have gained significant attention in modern biotechnology for manufacturing chemicals, leather goods, food, paper, pharmaceuticals, and textiles [14] and have further expanded into many novel areas such as functional foods, pharmaceuticals, and other health products that require exploration of novel sources of proteases [15].

Chia (*Salvia hispanica* L.) is an herbaceous plant that belongs to the order Lamiales, family Lamiaceae, subfamily Nepetoideae, and genus Salvia [16], with seeds comprising 25–40% oil, 17–24% protein, and 18–30% dietary fiber as major constituents [17]. Due to their high nutritional levels, including omega-3, calcium, iron [18], flavonoids, phenolic acids [19], and sterols [20,21], chia seeds have drawn attention for industrial applications with nutritionally enhanced functional food preparation. However, their industrial applications are limited to oil sources, and a large number of protein-rich by-products remain after oil extraction, which is often used to produce low-value-added products such as chia seed meal or animal feedstuffs [22]. Bioactive peptides are defined as specific protein fragments with relatively short peptide residue lengths (e.g., 2–9 amino acids), usually produced by microbial fermentation [23], enzymatic hydrolysis, and chemical hydrolysis [24], and have been proven to exhibit diverse antihypotensive, hypocholesterolemic, anticancer, immunomodulatory, and antibacterial activities [25]. Thus, considerable efforts have been made to obtain bioactive peptides from diverse protein-rich sources, such as milk proteins [26]. Recently, many researchers have reported the nutritional and functional properties of chia seed protein and its active peptides, including antihypertensive [27,28], anti-oxidative [27,29], anti-inflammatory [30], and cholesterol synthesis properties [31]. The peptide fraction showed higher antimicrobial activity than chia seed hydrolysates and helped to keep microorganisms at bay, making them a useful food additive [32]. Although several studies have shown that plant-based materials may be useful for the treatment and prevention of Alzheimer’s disease (AD) by inhibiting butyrylcholinesterase (BChE) and acetylcholinesterase (AChE), however, the effect of enzymatic hydrolysis of chia protein on AD has not been reported [33]. Only one recent study revealed that the high phenol content of chia seeds hinders the activity of AChE [34].

The aim of this study was to identify functionally active peptides from chia seed proteins using a novel protease obtained from novel *B. altitudinis* strains isolated from deep-sea mineral water in Korea. To this end, we first isolated protease-specific strains from deep-sea mineral water because of its extreme environment and unique ecosystems for various halophilic and pressure-resistant microorganisms. Then, we tried to apply the obtained crude protease to the extracted chia seed protein to find functionally active peptides and evaluated their biological properties, particularly their potential antioxidant activity against 1,1-diphenyl-2-picrylhydrazyl (DPPH) and 2,2′-Azino-bis (3-ethylbenzothiazoline-6-sulfonic acid) diammonium salt (ABTS), as well as the inhibition of AChE and BChE.

## 2. Materials and Methods

### 2.1. Chemicals and Media

*Bacillus* strains isolated from deep-sea mineral water were grown in de Man–Rogosa–Sharpe medium (BD, Detroit, MI, USA) consisting of 10 g/L protease peptone No. 3, 10 g/L of beef extract, 5 g/L of yeast extract, 20 g/L of dextrose, 1 g/L of polysorbate 80, 2 g/L of ammonium citrate, 5 g/L of sodium acetate, 0.1 g/L of MgSO_4_, 0.05 g/L of MnSO_4_ and 2 g/L of K_2_HPO_4_. Taq DNA polymerase (Gene All, Seoul, Republic of Korea) and TE buffer (10 mM Tris-HCl and 1 mM EDTA, pH8) were used for the polymerase chain reaction (PCR). Skim milk powder was purchased from Sigma Chemical Co. (St. Louis, MI, USA) to determine the proteolytic activity. All other chemicals and reagents used in this study were of analytical grade. Acetylthiocholine iodide, butyryl thiocholine chloride, diethylene tetramine pentaacetic acid (DTPA), AChE from Electrophorus electricus, 5,5′-dithiobis-(2-nitrobenzoic acid) (DTNB), BChE from equine serum, and pectinase from *Aspergillus aculeatus* were purchased from Sigma-Aldrich (St. Louis, MO, USA). All other chemicals and solvents were purchased from Merck (Burlington, MA, USA), Fluka (Buchs, Switzerland), and Sigma-Aldrich unless otherwise stated.

### 2.2. Isolation, Screening, and Culture Condition

Deep-sea mineral water obtained from the Geumjin area (Kang Neung, Kangwon-do, Republic of Korea) was used as the source for bacterial isolation after concentration by vacuum filtration. In brief, about 300 mL of hot spring water was filtrated using a mixed cellulose ester membrane filter of 47 mm in diameter (pore size, 0.2 μm; ADVANTEC; Toyo Roshi Kaisha, Ltd., Tokyo, Japan) held in a filter holder (Millipore Crop., Bedford, MA, USA) under vacuum. The obtained membrane filter paper was washed by shaking after the addition of 30 mL of hot spring mineral water in a Falcon tube. After concentration aseptically about 10 times, the deep hot spring mineral water was spread onto de Man–Rogosa–Sharpe (MRS) medium (Difco, Franklin Lakes, NJ, USA) with 1.5% Bacto Agar (Difco) and incubated at 37 °C for 3 days. Stock cultures of the isolates were stored in 30% glycerol at −80 °C. Prior to experiments, the isolates were sub-cultured from the frozen stokes on MRS broth at 37 °C with 180 rpm shaking and used as the activated condition.

### 2.3. Identification of the Bacillus Strains from Deep Hot Spring Sea Water

Sequence analysis was performed using an automated DNA sequencer (Applied Biosystems, Foster City, CA, USA), and sequence similarity analysis was conducted using BLAST (http://www.ncbi.nlm.nih.gov/BLAST/ accessed on 5 January 2024). To verify identities and demonstrate evolutionary relationships between Bacillus isolates, the 16S rRNA, *recA*, and *atpD* gene sequences, including necessary reference and type strain sequences retrieved from nucleotide databases, were independently subjected to phylogenetic analysis using MEGA. Concatenated gene trees were constructed with individual alignments of *atpD* and *recA* sequences using MEGA 6 software based on the UPGMA method. For detailed information on the primers used for PCR universal primer, amplifying *atpD* and *recA*, refer to Appendix A. A phylogenetic tree was constructed using TreeView (Win32). For further identification of the strains and sugar utilization, a commercially available biochemical test kit (API 50CHB kit; BioMérieux, Craponne, France) was used. For biochemical identification, the API 50CHB (*Bacillus*) system was used, following the modified method of Logan and Berkeley [35]. The results were scored according to the manufacturer’s instructions. A test scoring positive regardless of time was considered positive as some tests appeared positive at the first reading but reverted to negative by the time of the final reading. The assimilation pattern results were interpreted and identified using the analytical profile index [36] database in API web software (https://apiweb.biomerieux.com/ accessed on 5 January 2024).

### 2.4. Protease Activity of B. altitudnis 5-DSW

Protease activity was determined using a plate assay with a skim milk medium containing 1% (*w*/*v*) skim milk powder (Sigma-Aldrich) and 2% (*w*/*v*) Bacto agar [37,38]. The culture broth was centrifuged at 4 °C, 12,000× *g* for 10 min to remove insoluble cells. The supernatant was filtered through a 0.2 µm pore size membrane filter and then used as a crude enzyme solution. A 50 µL of crude enzyme solution was dropped onto 8 mm filter paper discs in the 1% skim milk plate. After incubation at 37 °C for 16 h, the diameter of the clear zone around the paper discs was measured. Relative activities are expressed as a percentage of the maximum protease activity. The protease activity of the crude extract was assayed according to Anson, 1938 using casein as the substrate [39]. Protease activity was calculated as follows:

Volume activity (Units/mL Enzyme) = (µmol tyrosine equivalents released) × (Total volume (in milliliters) of assay)/(Volume of Enzyme (in milliliters) of enzyme used) × (Time of assay (in minutes) as per the Unit definition) × (Volume (in milliliters) used in Colorimetric Determination), Weight activity (Units/mg solid) = (units/mL enzyme)/(mg solid/mL enzyme).

### 2.5. Optimization of Protease Activity from B. altitudinis5-DSW

The effects of carbon sources (10 g/L, *w*/*v*) and nitrogen sources on protease activity were examined with the basal medium (5 g/L yeast extract, 0.1 g/L K_2_HPO_4_, 0.1 g/L KH_2_PO_4_, and 1 g/L CaCl_2_, *w*/*v*) by changing glucose to lactose, fructose, galactose, or sucrose for carbon sources, or ammonium citrate, ammonium nitrate, ammonium sulfate, meat extract, peptone, polypeptide, and sodium nitrate for nitrogen sources. The effects of temperature and pH on protease activity, as well as the effects of medium components, were also evaluated. For optimal temperature, the crude enzyme was incubated at 20, 30, 40, 50, 60, and 70 °C for 60 min, and the absorbance was measured at 280 nm. The effect of pH on protease activity was determined by incubating the reaction mixture at pH ranging from 7.0 to 11.0 using different buffer systems (0.1 M): K_2_HPO_4_-NaOH (pH 7.0–8.0), Glycine-NaOH (pH 9–10), Na_2_HPO_4_-NaOH (pH 11). The protease activity was determined by measuring the residual activity after 2 h of incubation in the buffers of pH 7.0 to 11.0 at 40 °C.

### 2.6. Preparation of Enzymatic Chia Seed Protein Hydrolysates and Characterization of Hydrolysis Products

#### 2.6.1. Preparation of Enzymatic Chia Seed Protein Hydrolysates

Mucilaginous gels are mostly composed of soluble fiber [27]. Chia seeds were soaked in water (1:20, *w*/*v*) for 2 h and treated with pectinase (1% *w*/*w*, *A. aculeatus*) for 12 h. After washing 2 to 3 times with ultrapure water, the obtained chia seed crude extract was freeze-dried and defatted with n-hexane (1:6, *w*/*v*) at 45 °C for 75 min 3 times. The obtained defatted chia seed powder was suspended in deionized water (5%, *w*/*v*) and adjusted to pH 5 using 2 N HCl. After centrifugation at 37 °C, 100 rpm for 4 h, the samples were then adjusted to pH 10.0 by 2 N NaOH, followed by an additional 40 min centrifugation. After two more extractions, the collected supernatants were adjusted to pH 4.2 with 2 N HCl and allowed to stand at 4 °C for 16 h to precipitate the proteins. The proteins were separated by centrifugation at 18,000× *g* for 10 min, washed 3 times with acidic water (pH 4.2), and freeze-dried for further experiments. The obtained freeze-dried chia seed protein was suspended in a pH 9.5, 0.1 m sodium phosphate buffer solution (1%, *w*/*v*, dry basis) and mixed with crude protease solution from *B. altitudinis* 5-DSW (5 U mL^−1^) and incubated at 40 °C and 100 rpm for 2 h. After heating at 95 °C for 15 min, the insoluble portion was removed by centrifugation at 13,000× *g* for 15 min. The supernatant was then cooled to 4 °C for use for further study.

#### 2.6.2. Molecular Weight Analysis by Sodium Dodecyl Sulfate-Polyacrylamide Gel Electrophoresis and Size Exclusion Chromatography

For sodium dodecyl sulfate agarose gel electrophoresis analysis (SDS-PAGE), 30 µg of sample protein was mixed with sample buffer (Tris 0.125 M, SDS 4%, 2-mercaptoethanol 10% *v*/*v*, sucrose 10% *v*/*v*, bromophenol blue 0.004% p/v, pH 6.8) and incubated 5 min at 95 °C. The samples were then denatured using a 4% stacking gel and 12% separating gel. After electrophoresis, gels were stained with 0.25% Coomassie Brilliant Blue R-250. To assay the molecular weight distribution of the hydrolysate, fast protein liquid chromatography (FPLC) (BioRad, Hercules, CA, USA) equipped with a Superdex Peptide 10/300 GL column was used with the mobile phase with 50 mM Tris-HCl buffer pH 8, with 0.2 M NaCl, at a flow rate of 0.8 mL min^−1^. Proteins and peptides were detected by measuring the absorbance at 280 and 215 nm, respectively.

### 2.7. Antioxidant Activity Assays

#### 2.7.1. DPPH Radical Scavenging Activity

DPPH free radical scavenging activity was measured according to Brand-Williams, Cuvelier, and Berset, with slight modifications, and adapted to a 48-well flat-bottom plate [40]. Measurements were performed in triplicate. DPPH radical scavenging activity was calculated using the following formula:DPPH radical scavenging effect (%)/% Inhibition = A_1_ − A_2_/A_1_ × 100 (1)
where A_1_ is the absorbance of the control without the sample, and A_2_ is the absorbance in the presence of the sample and DPPH.

#### 2.7.2. ABTS^+^ Radical Scavenging Activity

The ABTS^+^ radical scavenging activity was determined according to Pukalskas et al., 2002 using a 48-well flat-bottom plate [41]. Measurements were performed in triplicate. The ABTS+ radical-scavenging activity was calculated using the following formula:ABTS^+^ radical scavenging effect (%)/% Inhibition = A_1_ − A_2_/A_0_ × 100(2)
where A_0_ is the absorbance of the control without the sample, A_1_ is the absorbance in the presence of the sample and ABTS^+^, and A_2_ is the absorbance of the sample blank without ABTS^+^.

#### 2.7.3. Inhibition of Acetylcholinesterase and Butyrylcholinesterase Activities

AChE and BChE inhibition assays were performed according to Ellman’s improved photometric detection method described previously [42], which is based on the reaction of the released thiocholine with a chromogenic reagent to give a colored product. Shortly, AChE solution, BChE solution, Acetylthiocholine iodide solution and DTNB solution were all prepared with 0.1 mol/L pH 8 phosphate buffer solution. A total of 150 μL of the 0.1 mol/L pH 8 phosphate buffer solution, 20 μL of AChE (BChE) with an enzyme activity concentration of 1.5 U/mL, 50 μL of 756 μmol/L DTNB were added in a 96-well plate, and finally 30 μL of sample solution at an appropriate dilution concentration were added as well. After the mixture was shaken well, it was placed at 37 °C for 5 min, then measured with a 96-well microplate reader, and the IC_50_ was calculated. A blank sample was prepared by adding buffer instead of the enzyme. Finally, inhibition was calculated by comparing the reaction rate in the test solution with that in the negative control. The experiment was performed in triplicate, and the inhibition was expressed as a percentage as follows:% inhibition = [1 − (A_1_/A_2_)] × 100 (3)

The results are expressed as the concentration of extract (IC_50_) in µg/mL that inhibited 50% of acetylcholinesterase, where A_1_ and A_2_ are the absorbance values of the enzyme activity with and without the samples, respectively.

## 3. Results and Discussion

### 3.1. Isolation of Strains with Protease Activity from Deep Sea Mineral Water

A deep-sea mineral water sample (300 mL) was aseptically obtained from 1100 m below the Kumjin reservation area (Gangwon, Republic of Korea) and concentrated 20-fold using sterilized filter paper. After cultivation, six yeast-like strains and eleven bacterial colonies (designated as DSW; deep-sea water) were isolated. Of these, only four isolates (1, 5, 8, and 11-DSW) exhibited relatively high protease activities, with strain 5-DSW showing the highest protease activity, as presented in Figure 1C.

The 5-DSW isolate formed circular, pearlescent, rough, and sticky colonies and was identified as a Gram-positive bacterium with slow motility and straight, rod-shaped cells measuring 2.0–6.0 μm in length, dividing by binary fission (Figure 2B). A biochemical analysis using the API 50CHB system revealed a 99.9% similarity to *Bacillus pumilus*. However, when the 16S rDNA gene sequence of 5-DSW, which displayed high protease activity, was analyzed through the GenBank database, it showed 100% similarity to *B. altitudinis, B. pumilus*, and *B. subtilis* et al. (Appendix A).

Despite the use of 16S rDNA gene sequencing, the gene alone was insufficient for differentiating *B. altitudinis* from *B. pumilus* and *B. subtilis*, all of which displayed 100% similarity. Therefore, distinguishing these isolates based solely on rRNA gene sequences proved cumbersome and, in some cases, inconclusive. To address this issue, two additional housekeeping genes, *recA*, and *atpD*, were employed and concatenated with the 16S rDNA sequences. The *recA* gene serves as an alternative molecular chronometer and is particularly useful for differentiating bacterial species with highly similar rRNA genes. Additionally, the *atpD* gene, which encodes the beta subunit of ATP synthase, is another widely used genetic marker that provides finer resolution for distinguishing between closely related bacterial species [43].

A phylogenetic tree was constructed based on the *recA* and *atpD* sequences of the Bacillus isolates, along with relevant reference strains retrieved from the GenBank database (http://www.ncbi.nlm.nih.gov/blast) (Figure 2). The comparison of these isolates with type strains and reference sequences revealed that the *recA* and *atpD* sequences of the isolates exhibited 100% similarity to *B. altitudinis*. In contrast, significant divergence was observed between these isolates and *B. pumilus* and *B. subtilis*, thereby confirming the identity of the strain as *B.altitudinis* 5-DSW.

### 3.2. Effects of Temperature, Carbon, and Nitrogen Sources on Protease Activity

The effects of carbon and nitrogen sources on protease activity were evaluated since protease activity is commonly affected by various components of the medium [44]. The protease production of microorganisms differs, particularly depending on the carbon and nitrogen sources in the medium. Skim milk agar plates were used to determine the effects of carbon and nitrogen sources on the extracellular protease activity observed in the medium. As shown in Figure 1A, none of the carbon sources affected protease production by *B. altitudinis* 5-DSW. When the effect of the nitrogen source was examined, *B. altitudinis* 5-DSW cultivated with meat extract showed the highest protease activity, followed by yeast extract (Figure 1B). Although ammonium nitrate and ammonium sulfate did not induce protease activity, ammonium citrate induced protease activity in this strain. This result is in line with the observation that ammonium represses enzyme production [45]. The optimum temperature on protease activity of *B. altitudinis* 5-DSW was relatively high at 40 °C (Figure 1D). Unexpectedly, the protease activity of *B. altitudinis* 5-DSW remained at approximately 70% of the predicted activity even at 70 °C. The protease activity of *B. altitudinis* 5-DSW showed high at an optimum pH of 9 and remained stable up to pH 11 (Figure 1D). The protease activity of the crude extract *B. altitudinis* 5-DSW was 460 U/mg casein.

### 3.3. Isolation of Chia Seed Protein Hydrolysate by Crude Protease from B. altitudinis 5-DSW

To obtain whole protein from chia seeds, abundant water-swollen gel components containing polymers of glucose (19.6%), galactose (6.1%), arabinose (9.6%), xylose (38.5%), galacturonic acid (5.3%), and glucuronic acid (18.7%) were first removed via pectinase treatment from *A. aculeatus* (1%, *w*/*w*) as described in Pedrolli, Danielle Biscaro, et al., 2009 [46]. The defatting of the expeller was carried out three times with n-hexane (1:6, *w*/*v*) at 45 °C for 75 min each time according to Xue-li et al., 2018 [47]. as shown in Table 1, After removing gel and oil components from chia seed sources, defatted chia seeds displayed a notable crude protein content of 11.3 g/100 g (dry weight), alongside a substantial dietary fiber content of 46.0 g/100 g (dry weight), and a reduced crude fat content of 0.93 g/100 g (dry weight). After freeze-drying, the crude protein obtained from the chia seeds in this study was almost 70.8%, which was higher than that reported by Xue-li et al., 2018 [47], who reported a yield of 61.52%. Generally, the protein extraction yield from chia seeds is comparatively higher than that from cottonseeds, sesame, peanuts, and soybeans because of abundant high-swelling mucilage gel formation with water, which hinders the ability of the protein to interact with the alkali solution [48]. Enzymatic hydrolysis by pectinase is useful for increasing the extraction rate of crude protein by lowering the degradation of galacturonic acid in chia seed mucilage saccharides.

When the crude chia seed protein obtained was enzymatically hydrolyzed by the crude protease from *B. altitudinis* 5-DSW, the obtained chia seed protein hydrolysates exhibited significantly increased DPPH radical scavenging activity after hydrolysis for over 2 h, which was approximately three times higher than that of the undigested chia seed protein, as shown in Figure 3A. The ABTS radical scavenging activity showed an identical pattern, but the enhanced activity was less than that of the DPPH radical scavenging activity. The chia seed protein hydrolysates obtained in this study showed significantly enhanced functional antioxidant activities, suggesting that they may affect the molecular structure, hindering their ability to serve as electron donors [28]. Although DPPH activity reached its highest value and remained stable after 2 h of hydrolysis, ABTS activity continued to increase until 6 h of hydrolysis, which is similar to the results of Sangsukiam et al., 2019 [49], as shown in Figure 4B. We selected a 6 h hydrolysate of chia seeds by crude protease from *B. altitudinis* 5-DSW and analyzed it via size exclusion chromatography and SDS-PAGE, as shown in Figure 3, to understand its degradation. As depicted in Figure 3A, a substantial alteration in the protein pattern was apparent after 6 h of hydrolysis compared with the undigested chia seed protein. Specifically, the electrophoretic profile of proteins with a molecular mass of approximately 50 kDa within the crude extract was almost completely eliminated after 6 h of hydrolysis (a), whereas the protein profile of approximately 30 kDa exhibited a marked reduction (b). Notably, the hydrolysates exhibited significantly expanded discernible bands in the lower molecular weight range (c), confirming the efficacy of the hydrolysis procedure. These results are in good agreement with those of the size exclusion chromatography, as shown in Figure 4B. However, significant amounts of high molecular weight proteins were observed in the hydrolysate, as shown in Figure 3B. The lower molecular weight protein hydrolysate increased after treatment with crude protease from *B. altitudinis* 5-DSW. Our results clearly show that the enhanced functional properties of chia seed hydrolysate obtained from the hydrolysis of crude protease from *B. altitudinis* 5-DSW must be effective in changing the protein profile by enhancing lower molecular weight proteins, resulting in enhanced antioxidant activities.

### 3.4. Acetylcholinesterase and butyrylcholinesterase Inhibitory Activities of Chia Seed Protein Hydrolysate

Acetylcholine and butyrylcholine deficiencies are highly associated with memory impairment in Alzheimer’s disease, and their regulation is highly correlated with AChE and BChE, respectively [28]. In the present study, as shown in Table 2, the obtained chia seed protein hydrolysate by crude protease of *B. altitudnis* 5-DSW showed hydrolysis time-dependent inhibitory effects on AChE activity with an IC_50_ value of 14.48 ± 0.88 μg/mL for the 6 h hydrolysate, followed by 8 h (IC_50_ value of 14.57 ± 0.59 μg/mL) and 4 h (IC_50_ value of 20.69 ± 0.98 μg/mL), respectively, which were 2 to 3 times higher than the non-digested chia seed protein with an IC_50_ value of 59.06 ± 0.25 μg/mL. The obtained hydrolysate peptide for 6 h also exhibited BChE inhibition activity with the most potent inhibitory activity IC_50_ value of 10.90 ± 0.80 μg/mL, followed by 8 h (IC_50_ value of 11.20 ± 0.81 μg/mL) and 4 h (IC_50_ value of 12.30 ± 0.91 μg/mL) whereas chia seed protein without hydrolysate showed poor inhibitory activity against BChE. Generally, the hydrolysis of proteins derived from plants or seeds shows enhanced functional activities on AChE, with a decreased degree of hydrolysis of high-molecular-weight proteins [50]. When we compared the AChE or BChE activities with those of other food source proteins, we found high extinguishable activities [51]. Considering that bioactive protein hydrolysates derived from chia seeds could have a positive influence on human health without the undesirable negative side effects associated with drugs during long-term use, our results represent a milestone for developing promising natural therapeutic and preventive agents with minimal toxicity and side effects against oxidative stress and neurodegenerative diseases, such as AD [52].

## 4. Conclusions

Although chia seeds have gained growing significance in human health owing to their high nutritional composition, their use is currently limited by their own properties, without any additional food technology applications. In this study, we attempted to obtain a nutritionally enhanced protein hydrolysate from chia seed protein using a novel protease directly obtained from a newly isolated protease-producing strain, *B. altitudinis* 5-DSW, from deep hot spring mineral water. This strain exhibited significant protease activity even at high temperatures, indicating its potential for industrial applications. The crude protease obtained from *B. altitudinis* 5-DSW can effectively hydrolyze chia protein and shows high antioxidant and inhibitory activities against AChEs. These findings suggest that the proteases from *B. altitudinis* 5-DSW and chia seed protein hydrolysates have potential applications in biotechnology and functional food development. Future research is essential to isolate and identify these biopeptides and assess their potential for incorporation as dietary supplements in various food matrices.

## Figures and Tables

**Figure 1 microorganisms-12-02048-f001:**
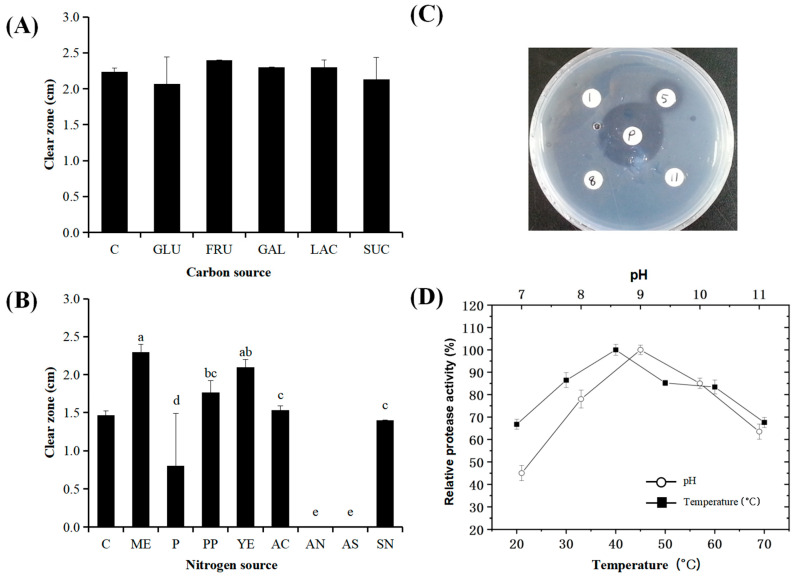
(**A**) Effect of carbon sources on protease activity of crude enzyme obtained from *B. altitudinis* 5-DSW; C, effect of crude enzyme produced with the basal medium, GLU, glucose, FRU, fructose, GAL, galactose, LAC, lactose, SUC, sucrose; All strain did not show significant differences (*p* < 0.05) by ANOVA with Duncan’s multiple range test. Results were expressed with mean ± SD of triplicate experiments. (**B**) Effect of nitrogen sources on protease activity of crude enzyme obtained from *B. altitudinis* 5-DSW; C, effect of crude enzyme produced with the basal medium, ME, meat extract, P, peptone, PP, polypeptone, YE, yeast extract, AC, ammonium citrate, AN, ammonium nitrate, AS, ammonium sulfate, SN, sodium nitrate, Values with different superscripts are significantly different (*p* < 0.05). Standard deviation bars were indicated, and results were expressed with mean ± SD of triplicate experiments. (**C**) Fibrinolytic activity of *B. altitudinis* 5-DSW isolated from deep-sea mineral water. The numbers 1, 5, 8, and 11 indicate different strains of *B. altitudinis* 1-DSW, 5-DSW, 8-DSW, and B. megaterium 11-DSW) used in the experiment, while “P” denotes the presence of protease activity. (**D**) Effect of temperature and pH on relative protease activity of the crude enzyme obtained from *B. altitudinis* 5-DSW. Results were expressed with mean ± SD of triplicate experiments.

**Figure 2 microorganisms-12-02048-f002:**
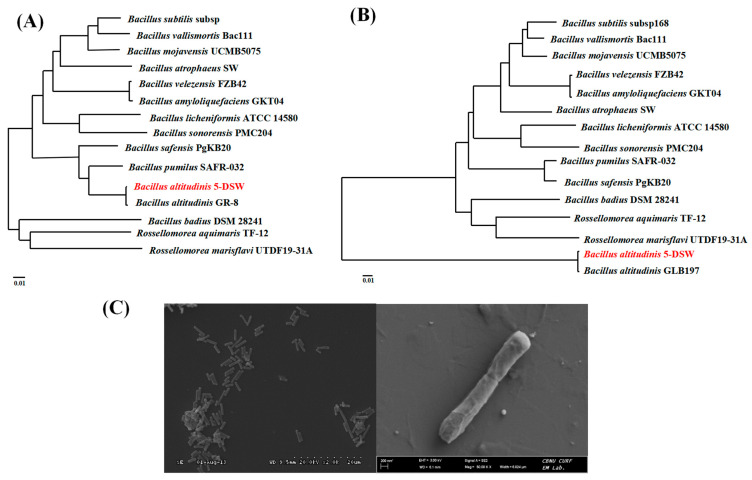
(**A**) The *recA* phylogenetic tree; (**B**) The *atpD* phylogenetic tree. The evolutionary history was inferred using the UPGMA method. The percentages of replicate trees in which the associated taxa clustered together in the bootstrap test (1000 replicates) are shown next to the branches. The tree is drawn to scale, with branch lengths in the same units as the evolutionary distances used to infer the phylogenetic tree. The evolutionary distances were calculated using the p-distance method and are expressed as the number of base differences per site. (**C**) Scanned electron microscopic (SEM) image of the *B. altitudinis* 5-DSW strain.

**Figure 3 microorganisms-12-02048-f003:**
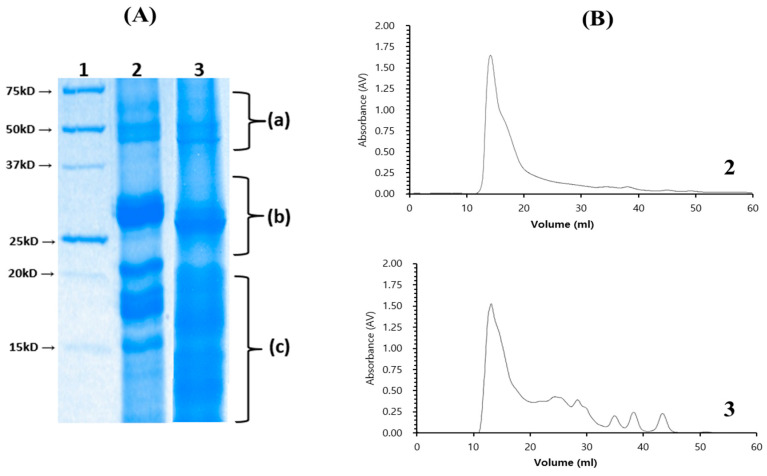
(**A**) SDS-PAGE map of chia seed protein stained with Coomassie Brilliant Blue. Comparative molecular weight fractions are made with curly braces (a), (b), and (c); (**B**) Size exclusion chromatography using a Superdex Peptide 10/300 GL column, monitored at 280 nm. Molecular weight standards were confirmed using the Gel Filtration LMW Calibration Kit (Cytiva, Marlborough, UK, 28403841). 1: Marker: The marker was obtained from Bio-Rad Laboratories (Catalog No. 161-0363). 2: chia seed protein. 3: 6 h hydrolysate chia seed protein.

**Figure 4 microorganisms-12-02048-f004:**
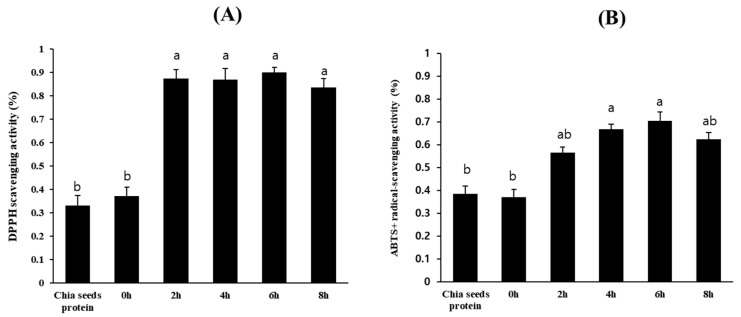
Effect of hydrolysis time on antioxidant abilities; (**A**) scavenging of scavenging of 2,2-diphenyl-1-picrylhydrazyl (DPPH) radicals; (**B**) 2,2′-azinobis (3-ethylbenzothiazoline-6-sulfonic acid) radical cation (ABTS) cation ion free radicals. Values with different superscripts are significantly different (*p* < 0.05). Results were expressed with mean ± SD of triplicate experiments.

**Table 1 microorganisms-12-02048-t001:** Defatted Chia Seeds Ingredients g/100g dbw, mean ± S.D., *n* = 3.

Chia Seed		Crude Protein (g)	Crude Fat (g)	Protein Recovery Yield (%)
Step 1	Source	15.4 ± 0.48	29.1 ± 0.63	100
Step 2	Defatted process	11.3 ± 0.75	0.93 ± 0.3	73.37
Step 3	Freeze-dried process	10.9 ± 0.9	0.90 ± 0.3	70.77

Source of chia seeds: country of Paraguay, Subdivision sales as Hello green; Ingredients g/100 g dbw, mean ± S.D., *n* = 3.

**Table 2 microorganisms-12-02048-t002:** ChEs inhibitory activities of the chia seed protein after hydrolysis.

Chia seeds Hydrolysis	AChE IC_50_ (μg/mL)	BChE IC_50_ (μg/mL)
No hydrolysis	59.06 ± 0.25	45.48 ± 0.88
0 h	58.67 ± 0.30	43.27 ± 0.51
2 h	26.17 ± 0.35	24.57 ± 0.59
4 h	20.69 ± 0.98	12.30 ± 0.91
6 h	14.48 ± 0.88	10.90 ± 0.80
8 h	14.57 ± 0.59	11.20 ± 0.81

## Data Availability

The original contributions presented in the study are included in the article/Appendix A, further inquiries can be directed to the corresponding author.

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
