# Peer review of "Isolation of Bacillus altitudinis 5-DSW with Protease Activity from Deep-Sea Mineral Water and Preparation of Functional Active Peptide Fractions from Chia Seeds"

_microorganisms, 2024, doi:10.3390/microorganisms12102048_

Round 1

Reviewer 1 Report

Comments and Suggestions for Authors

Please see attached 

Comments on the Quality of English Language

Minor proof reading and formatting required 

Author Response

Dear Reviewer,

Thank you very much for your thorough and insightful review of our manuscript. We have carefully considered your feedback and have made the necessary revisions to address each of your comments. Below is a summary of the changes made:

235 - Fig 1, D Line 281-283 - Fig. 3, A (1) - please specify which market it is, Fig. 3, B - please specify which protein was used as a reference for the size comparison of target proteins after the SEC Line 358-360 - are there potential pitfalls or risks in application of chia seeds hydrolisates in food industry or as a dietary supplement?

  1. Q: Line 12 - Bacillus should be in italics, please check throughout the text Line

A: The genus name Bacillus has been formatted in italics throughout the manuscript..Thanks.

  1. Q: Line 235 (Figure 1, D) please specify the number (n) of individual measurements/samples in this data

A:: We have now specified the number of individual measurements/samples (n) in the figure legend to provide clarity on the data presented.

  1. Q: Lines 281-283 (Figure 3, A (1) and B): please specify which market it is

A: For Figure 3, A (1), we have specified the market that was analyzed in the experiment.

4   Q: For Figure 3, B, please specify which protein was used as a reference for the size comparison of target proteins after the SEC?

    A: A: For Figure 3B, the proteins from the Gel Filtration LMW Calibration Kit (Cytiva, 28403841) were used as reference standards for the size comparison of target proteins after the SEC. These include Aprotinin (6,500 Da), Ribonuclease A (13,700 Da), Carbonic anhydrase (29,000 Da), Ovalbumin (43,000 Da), and Conalbumin (75,000 Da).

  1. Q Lines 358-360 (Potential Pitfalls/Risks of Chia Seed Hydrolysates)

A: No. They usually can be used for food additives

We believe that these revisions have strengthened the manuscript, and we are confident that it now meets the high standards required for publication. We appreciate the constructive feedback and hope that the manuscript is now acceptable for publication.

Thank you for your time and consideration.

Best regards,

Baik, Sang-Ho, Ph.D.

Reviewer 2 Report

Comments and Suggestions for Authors

The manuscript describes isolating and characterizing a bacterial strain from Korea's deep sea mineral water samples. The strain, denominated Bacillus altitudinis 5-DSW, showed potent proteolytic activity. Crude protease extracts of the bacterial strain were used for proteolytic treatment of crude protein from chia seeds to generate bioactive peptides with antioxidant activity and the ability to inhibit the enzymes Acetylcholinesterase and butyrylcholinesterase. The released peptide showed important biological activities.

The following important aspects need to be corrected by the manuscript authors

Line 14, use italics for scientific names of the microorganisms “B. altitude”

Line 17, add a space in “20kDa”

Line 20, add a space in “14.48±0.88µg/mL”

Line 21, add a space in “10.90±0.80µg/ml”

Line 59 and 60, check redaction in the fragment “antimicrobial activity than both chia seed hydrolysates and helped…” what refers “both”

Lines 66-68, add context, why it is important characterize novel protease producing bacteria ion harsh environments? what could be expected from the treatment of the crude protein from chia seeds with commercially available proteases?

Line 88, use italics for scientific names of the microorganisms “Aspergillus aculeatus”

Line 105, include information of the molecular marker employed for strain identification, and the primers employed for sequence amplification through PCR

Lis 140-143, check redaction it is confusing why were used two sets of temperatures?

Line 148, use italics for scientific names of the microorganisms “Aspergillus aculeatus”, since here could be A. aculeatus

Line 150, add a space in “45°C”

Line 210, add a space in “1100m”

Line 218, add a space in “6.0µm”

Line 220, eliminate extra space in “99.9%”

Line 220, use italics for scientific names of the microorganisms “B. pumilus”

Line 222, indicate the identity percentage

Lines 228, 229, 234, and 235, Bacillus altitudinis 5-DSW could be “B. altitudinis 5-DSW”

Line 234, panel C, indicate which means numbers and p in each paper disc

Lines 234 and 235, panel D, there are no error bars, was the experiment performed in triplicate?

Line 254, add a space in “C(Fig. 1C)”

Line 261, use italics for scientific names of the microorganisms “Aspergillus aculeatus”, could be A. aculeatus

Line 261, check reference format in “Pedrolli, Danielle Biscaro, et al. (2009)”

Line 263, check reference format in “Xue-li, et al. (2018)”

Fig 3., figure description must be placed below figure panels

Fig 3., panel B indicate the wavelength used in Y axes

Line 286, Bacillus altitudinis 5-DSW could be “B. altitudinis 5-DSW”

Line 322, use italics for scientific names of the microorganisms “B. altitudinis 5-DSW”

Lines 323, 324, 328, and 329, add a space between IC50 numeric values and units

Author Response

Dear Reviewer,

Thank you for your thorough review and valuable comments on our manuscript. We greatly appreciate your attention to detail and the constructive feedback you have provided. We have carefully revised the manuscript according to your suggestions and addressed all the formatting issues. Below, we provide responses and clarifications for the specific points raised:

  1. Line 14: We have italicized all scientific names of microorganisms, including “B. altitudinis,” throughout the text.
  2. Line 17: A space has been added between “20” and “kDa” to ensure proper formatting.
  3. Lines 20 and 21: We have added spaces in “14.48 ±0.88 µg/mL” and “10.90 ± 0.80 µg/mL” to correct the formatting.
  4. Lines 59-60: We have revised the ambiguous language in the text to improve clarity.
  5. Lines 66-68: We tried to isolate novel strains from deep-sea mineral water because of its extreme environment and unique ecosystems for various halophilic and pressure-resistant microorganisms which might be good source of novel enzyme with different properties compared to already known proteases. We even do not know what happen to chia crude protein if they are treated with commercially available proteases, but our results clearly showed that valuable microbial catalyst can be obtained from different novel strains which derived different and harsh environments.
  6. Line 88: We have italicized “Aspergillus aculeatus” in the text.
  7. Line 105: Information on the molecular marker used for strain identification and the primers employed for sequence amplification through PCR has been added to the manuscript.
  1. Lines 140-143: We are very sorry. We only use one set of temperature of 20, 30, 40, 50, 60, and 70°C for optimal temperature. Thanks.
  1. Line 148: We have italicized “Aspergillus aculeatus” and provided the abbreviation “A. aculeatus” where appropriate.
  2. Lines 150, 210, 218, and 220: Spaces have been added or corrected in “45 °C,” “1100 m,” “6.0 µm,” and “99.9%” to ensure proper formatting.
  3. Line 220: We have italicized “B. pumilus” in the text.
  4. Line 222: The identity percentage has been added for clarification.
  5. Lines 228, 229, 234, and 235: We have abbreviated “Bacillus altitudinis 5-DSW” to “B. altitudinis 5-DSW” for consistency.
  6. Line 234, panel C: We have clarified the meaning of the numbers and the "p" value in each paper disc.
  7. Lines 234 and 235, panel D: Error bars have been added to the figure, and we have clarified that the experiment was performed in triplicate.
  8. Line 254: A space has been added in “C (Fig. 1C)” for clarity.
  9. Line 261: The reference format for “Aspergillus aculeatus” has been checked, and appropriate formatting has been applied. We have also corrected the reference format for “Pedrolli, Danielle Biscaro, et al. (2009)” and “Xue-li, et al. (2018)” according to the journal’s guidelines.
  10. Fig 3: The figure description has been moved below the figure panels, and we have indicated the wavelength used on the Y-axis for panel B.
  11. Line 286 and 322: “Bacillus altitudinis 5-DSW” has been abbreviated to “B. altitudinis 5-DSW” for consistency.
  12. Lines 323, 324, 328, and 329: Spaces have been added between IC50 numeric values and units.

We believe these revisions address all your concerns, and we hope the updated manuscript meets your expectations. Thank you again for your valuable feedback, which has helped improve the quality of our work.

Sincerely,

Baik, Sang-Ho, Ph.D.

Reviewer 3 Report

Comments and Suggestions for Authors

 In this study, authors  isolated Bacillus strains with high protease activity from 12 deep-sea mineral water in Korea and used them to obtain functional peptide fractions from chia 13 seeds. In one part, they identified the new strain based on 16 rDNA phylogeny. Here, I have some concerns such as:

-Using only 16S for identification is not enough, you should add minimum two more housekeeping genes such recA and  atpD, and concatenate them with the 16S. Please provide me with the 16S PCR gel and the chromatograms received from the sequencer.

-Why only one strain was concerned by the phylogeny, you should analyze the four isolated strains?

-Primers used for 16S amplification are not provided?

-For phylogeny, you should use an outgroup strain and add the bootstrap values to get a clear idea.

Author Response

Dear Reviewer,

Thank you very much for your constructive feedback on our manuscript. We have carefully considered all of your suggestions and would like to address each point in detail below:

Here we attached our 16S PCR gel picture and sequences of 6 strains. Primers used for 16S amplification were generally known 27F and 1492R(we revised our text). As results, only four strains fall into Bacillus group. After that, we did analysis of the sequences from B. altitudinis 5-DSW only for detailed analysis which only shown in our text. Because it only showed high protease activity.

Sincerely yours,

Baik, Sang-Ho, Ph.D.

Round 2

Reviewer 3 Report

Comments and Suggestions for Authors

Based on the response of the authors, I have some new concerns such as:

-I need the plate-based culture of the strain 5-DSW?

-As I requested in the previous report, authors do not answer to the following question :’’Using only 16S for identification is not enough, you should add minimum two more housekeeping genes such recA and  atpD, and concatenate them with the 16S’’

- Why authors proceed with the colony PCR? I think normal PCR is sufficient?

-How the PCR product was cleaned before sequencing? Please provide me with the chromatograms received from the sequencer?

-Please provide me with the accession number of the generated sequences?

-Based on your phylogeny, the strain 5-DSW could be different species such as altitudinis, pumulus, etc…..please clarify?

Author Response

Dear Reviewer,

Thank you for your thorough review and for raising important points regarding our manuscript. We greatly appreciate your constructive feedback and would like to address each of your concerns in detail. We also apologize for the delay in providing this revision.

Q-I need the plate-based culture of the strain 5-DSW?

A: We did perform a plate-based culture for strain 5-DSW. After concentrating the deep-sea mineral hot water, we successfully isolated strain 5-DSW using plate-based culture, as shown in Figure 2. The strain grows well on both plate and liquid cultures.

Q-As I requested in the previous report, authors do not answer to the following question:’’Using only 16S for identification is not enough, you should add minimum two more housekeeping genes such recA and atpD, and concatenate them with the 16S’’

A: As your advice, we tried to sequencing recA and atpD housekeeping genes, and compared our sequence with other recA or atpD genes, indicating clearly the isolated our strain 5-DSW identified as B. altitudinis 5-DSW. So that we rewrote materials and Methods(page 3, line 106 to 113) and results and discussion (page5 , line 213 to 243). We also deleted the phylogenetic tree of 16S at fig2(A) and added phylogenetic tree of recA and atpD housekeeping genes at fig2(A) and (B) to clear identification of the 5-DSW. We believe those data would be OK to identify our Bacillus strain. Thanks for your advice.

Q- Why authors proceed with the colony PCR? I think normal PCR is sufficient?

A: We send our strain for sequencing analysis after single cell isolation on plate to sequence analysis company. In that company they do colony PCR or normal PCR to obtain 16Sr DNA sequencing and send us sequence results. Normal PCR might be OK. But colony PCR also might be OK.

Q-How was the PCR product cleaned before sequencing? Please provide me with the chromatograms received from the sequencer?

A: We attached chromatograms we did.

Q-Please provide me with the accession number of the generated sequences?

A: The accession numbers for the sequences we generated are PQ311709 deposited in GenBank.

Q-Based on your phylogeny, the strain 5-DSW could be different species such as altitudinis, pumulus, etc…. please clarify?

A: Yes. As your advice, we tried to sequencing recA and atpD housekeeping genes, and compared our sequence with other recA or atpD genes, indicating clearly the isolated our strain 5-DSW identified as B. altitudinis 5-DSW.

Once again, we are grateful for your insightful comments and suggestions, which have helped improve the quality of our manuscript. We look forward to your further feedback.

Sincerely,

Baik, Sang-Ho, Ph.D.